# Dimension Mixer: Group Mixing of Input Dimensions for Efficient Function Approximation

Suman Sapkota[1],[*] Binod Bhattarai[2]
[1]NAAMII, Nepal [2]University of Aberdeen, UK
suman.sapkota@naamii.org.np, binod.bhattarai@abdn.ac.uk

The recent success of multiple neural architectures like CNNs, Transformers, and MLP-Mixers motivates us to look for similarities and differences between them. We find that these architectures can be interpreted through the lens of a general concept of dimension mixing. Research on coupling flows, shufflenet and the butterfly transform shows that partial and hierarchical signal mixing schemes are sufficient for efficient and expressive function approximation. In this work, we study group-wise sparse, non-linear, multi-layered and learnable mixing schemes of inputs and find that they are complementary to many standard neural architectures. Following our observations and drawing inspiration from the Fast Fourier Transform, we generalize Butterfly Structure to use non-linear mixer function allowing for MLP as mixing function called Butterfly MLP. We are also able to sparsely mix along sequence dimension for Transformer-based architectures called Butterfly Attention. Experiments on CIFAR and LRA datasets demonstrate that the proposed Non-Linear Butterfly Mixers are efficient and scale well when the host architectures are used as mixing function. We devise datasets with increasing complexity to solve Pathfinder-X task. Additionally, we propose Patch-Only MLP-Mixer for processing spatial 2D signals demonstrating a different dimension mixing strategy.

## 1. Introduction

The recent success of various Neural Network Architectures such as Convolutional Neural Network (CNN)[1–4], Transformers[5, 6], and MLP-Mixer[7] can be credited to the sparse structured processing of input signals and the function or parameter sharing used in these architectures. These architectures are suitable for many downstream tasks, such as image classification, while working differently from each other. Although they share common parts, such as processing in patches, and sharing processing functions across tokens, they also mix signals at different locations in a variety of ways depending on the architecture.

This abundance of processing the same input in multiple ways towards the same objective allows us to analyse and find common mechanisms among these architectures as well as differences as shown in Table 1. Moreover, these architectures are quadratic with respect to some of the input dimensions. For example, with sequence length in Attention, with channel dimension in Convolution or with patch/channel dimension in MLP-Mixer.

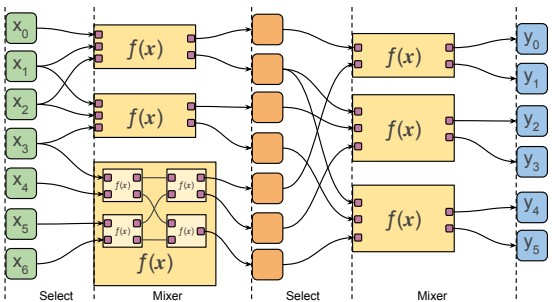

Figure 1: An example of a general Dimension Mixer model, split into multiple layers of (i) *Select* and (ii) *Mix* stages. The Select stage selects input dimensions for each of the Mixer units. The Mix stage processes inputs and is learned via optimization. This example shows that the mixers can use arbitrary dimensions and have varying capacity. The mixers can themselves be Dimension Mixer Model.

---

[*]Independent Researcher. Research primarily done while at NAAMII.

Second Conference on Parsimony and Learning (CPAL 2025).

Table 1: Overview of properties of various architectures.

| Model | Unit | Strictured | Input Sparsity | Mixing | Dense Operation |
|---|---|---|---|---|---|
| Linear (or LowRank) | dot-product | ✗ | None | 1-layer | - |
| Linear-Butterfly | linear | ✗ | block sparse | butterfly-factors | block-linear |
| MLP-Butterfly (ours) | MLP | ✗ | block sparse | butterfly-factors | block-MLP |
| CNN | convolution | ✓ | sliding window | radial | conv channels |
| MLP-Mixer | MLPs | ✓ | patch & channels | 1-block | channel+patch MLP |
| Patch Only MLP (ours) | MLPs | ✓ | patch | radial | patch MLP |
| Transformer | Attention & MLP | ✓ | tokens & sequence | 1 block | KQV, Attn & MLP |
| Butterfly Attn. Tranf. (ours) | Attention & MLP | ✓ | tokens & sequence | butterfly-factors | KQV & MLP |
| Dimension Mixer (Generalization) | Any complete mix $A \to B$ transform | Any | Group $\leq$ Total Dims | Mix all dims in $L$ layers | Any |

We search for a general sparse signal processing mechanism that allows to sparsify structured as well as unstructured dimensions of input tensor.

Processing structured data with neural networks relies on model's inductive biases, such as translation eqivariance with 2D images, leading to highly parallelizable and parameter efficient processing. However, for unstructured input dimensions, general dense transforms do not scale well with the dimensionality of the input signal. While sparse parametrizations can partially alleviate this issue [8], general sparse matrix multiplication algorithms are lagging in performance compared to dense counterparts. Low-rank matrix parameterizations reduce the complexity by applying associativity of dense low-rank components multiplication[9] and lead to better performance. Structured sparse matrices (butterfly matrices) as used in the Fast Fourier Transform (FFT) [10] and recently in neural networks [11–13] show great promise to parameterize dense matrices for efficient matrix multiplication.

Multilayered partial signal processing of unstructured input dimensions is also a recurring pattern of many deep architectures such as Coupling-Flows[14, 15] and ShuffleNet[16, 17]. Armed with these observations, we propose a general sparse signal processing model termed the Dimension Mixer model. It leverages group-wise mixing of input signals, processed in layer-wise fashion such that they communicate with each other, i.e. for $\mathbb{R}^N \to \mathbb{R}^M$ transformation, all $N$ input dimensions can have non-zero derivative with respect to all $M$ output dimensions. In other words, there must exist a mixing path from each input dimension to each output dimension. Dimension Mixer consists of several parallel dimension *Select* and *Mix* stages, as demonstrated in Fig. 1. This mechanism is repeated in a sequence of layers such that input dimensions can be mixed densely.

Inspired by the widely used sparse signal mixing method from the FFT, which uses butterfly matrices and block-wise processing, we generalize the butterfly structure beyond Linear Transform to Non-Linear mixing function. Since such mixing borrows efficiency and scalability from FFT, it has $\mathcal{O}(N \log_r N)$ complexity, where $N$ represents the number of dimensions and $r$ represents mixing block size ("Radix-$r$"). Although butterfly structures have been previously explored in both learned and non-learned settings to replace dense matrix multiplication, our primary contribution is the extension of this paradigm to non-linear setting while retaining the structured mixing of different dimensions of the input tensor. We propose an efficient Butterfly Attention Mixer, a block-structured self-attention mechanism, with sub-quadratic complexity in Sequence Length(S) which attends global tokens with later layers. We additionally devise a new mixing strategy for 2D images called Patch-Only MLP Mixer. It lies between the original MLP Mixer and a CNN; this view helps to unify the working mechanism of both architectures. To process 2D images, we propose use of only patch-level mixing strategy. We conduct studies of varying scale to evaluate efficiency and approximation capability of our methods in several datasets.

Experiments on use of sparse MultiLayer Perceptron (MLP) in MLP-Mixer architecture on CIFAR-10 dataset shows that Butterfly MLP produces better results than sparse Butterfly Linear. The experiments on Patch-Only MLP-Mixer shows that our architecture produces smaller hidden representation and allows for more computational efficient mixing of image signal.

Furthermore, we find that sparse Butterfly Attention is sufficient for CIFAR-10 and CIFAR-100 datasets, produces faster architectures and scales better for large sequence lengths as well. We largely credit the performance to structured nature of butterfly attention when used in image. Experiments on the Long Range Arena (LRA) Benchmark also demonstrate that the Butterfly Attention performs better than the baselines on the Retrieval, Image and Pathfinder tasks with best Average accuracy.

Finally, we demonstrate the ability of our method to solve the challenging Pathfinder-X task, with the sequence length of 16,384 tokens. We devise a continuous learning dataset from small pathfinder tasks to the Pathfinder-X to solve it efficiently. This task remained untackled by many low-rank attention methods; this attests to the generality and efficiency of our method.

## 2. Background and Related Works

**Structured Processing in Deep Learning:** One of the main ingredients to the success of deep learning frameworks is using structured processing units to handle structured data. These processing units enable the parallel processing of structured data, leveraging the power of accelerated computing technology such as GPUs and TPUs. The CNNs [1, 2, 4], in general, have sliding window filters which apply a linear transformation to patches of the image. From the perspective of signal mixing, the CNN architecture generally works by increasing the receptive field of filters as the number of layers increases. This allows the later layers to attend to a larger region of the input image, although indirectly through previous kernels. The CNN was the only dominant architecture for vision till the advent of Vision Transformers (ViT) [6]. ViT generally works with non-overlapping patches, and the mixing of the patches happens immediately by the attention layer without waiting for later layers to comprehend the whole input signal. It is followed by processing the signals per patch/token by the MLP layer. However, Vision Transformers lack a sliding window, thus preventing the shift equivariance inductive bias of CNNs. Contrary to this, MLP-Mixer [7] replaces an attention layer with a channel mixing layer which is equally effective.

ViT [6] scales poorly against large sequence lengths, thus preventing the division of images into a large number of patches. Later, Swin Transformer [18] and RegionViT[19] tackled such problems using convolutional priors and hierarchical attention. Majority of the works in transformers for efficiency [20] focus on creating sparse attention pattern [21–33]. While some of the works also focus on making the sparse MLP block [34].

Recently, the volume of literature in MLP-Mixer is also getting equally bigger[35–42]. Most of these studies investigate the different ways of processing the input signals such as gating-based mixing [43], shift in channels [44] or using FFT for mixing tokens [45]. The majority of the works in this category involve mixing either the patches and/or channels in various ways. These works motivate us to design a Patch-Only MLP-Mixer, which only mixes patch-wise for signal processing similar to CNN.

**Partial Signal Processing:** ShuffleNet [16] uses group convolution over the channel dimensions, allowing for efficient convolution due to the reduced number of channels. It also mixes the channels for evenly distributing the output signals to the next convolution layer. It is important to note that AlexNet[2] uses parallel grouped convolution for accelerating on two GPUs and combines the hidden states in some layers to process them jointly. Similarly, Megatron-LM[46] splits the input tokens into two blocks, process them independently and again combines them at the end of the attention and MLP blocks. Such a *split*, *process*, and *combine* method allows for processing a large number of tokens in a parallel and efficient manner.

Coupling Flows [14, 47–49] and Reversible ResNet [15], use split and process mechanism that enables invertibility. Partial mixing of signals can approximate any diffeomorphic function [50]. The key takeaway with these architectures is that partial signal processing is sufficient for function approximation and provides efficiency and scalability benefits. These works motivate our generalization to Dimension Mixer model.

**Sparse Linear and Non-Linear Models:** The complexity of matrix multiplication for a vector input is known to be $N^2$, where $N$ is the input-output dimension. This problem has been tackled to

some extent using Fast Matrix Multiplications [51, 52] and Low-Rank Matrix Decomposition. The low-rank transformation has been widely used in efficient CNN architectures [53, 54]. Models like EffecientNet-v2 [55] and MobileNets [56] use depth-wise and point-wise convolution as a low-rank factorization of a standard convolution. One can get a highly sparse matrix using pruning-based techniques [57]. However, these methods accelerate neural networks on CPUs and mobile devices but fail to accelerate significantly on GPUs.

**Linear Butterfly Sparsity:** The block sparse matrices [58] have been used widely to accelerate matrix multiplication. Another alternative is to use butterfly matrices [11, 59], which is inspired by the FFT and scale well with dimensions. Some of the past works [11–13, 60] have successfully used linear butterfly transformation to replace dense transformation and have produced highly efficient architectures. Furthermore, structure of butterfly sparsity is block sparse, thus highly parallelizable and have highly efficient hardware [60] and software implementations. In this paper, we generalize the Butterfly Sparsity to arbitrary non-linear dense operation.

**Block Sparsity in Attention:** Due to high parallelization of block-sparse operations, there has been developments of block-sparse attention patterns. Previous works focus on combining block-sparse with other global component like random [61] or low-rank [26]. Blockwise-Transformer [62] focuses on using various rotated block-sparsity patterns in different attention heads. Pixelated-Butterfly [12] mentions use of butterfly sparsity and low-rank to create approximate attention similar to [23].

**Long Range Arena (LRA):** LRA [63] is one of the most challenging benchmarks for evaluating the quality of a model in long-range sequences (more than 1K tokens) involving various types of problems. ListOps [64] consists of hierarchical mathematical operations which measure the parsing and analytical abilities. Byte-level text classification [65] is a binary classification task on characters. Byte-level document retrieval [66] measures the ability to create a compressed representation of input. Similarly, Image Classification Task [67] is on a sequence of flattened pixels. Pathfinder task, including the Pathfinder-X [68] is a synthetic dataset of labyrinths formulated as binary classification task. Succeeding on all of these tasks shows that an efficient Attention mechanism works on diverse and challenging tasks formatted as long sequence problems. The dataset provides a medium for fair comparison using similar hyperparameters, rather than a competition using different configurations.

## 3. Dimension Mixer Models

Dimension Mixer model is a simple and general structure observed among most deep learning architectures. The key observation is that of structured signal processing and performing mixing of all input signals efficiently. Computational efficiency is achieved thanks to partial (group-wise) mixing of input signals. For structured data, the mixing is itself structured similar to the data and mixing functions like MLPs can be shared. Parameter efficiency is gained due to function sharing as well as partial mixing.

Geometric Deep Learning [69, 70] tackles the generality of these architectures by considering graphs as an underlying data structure, however, we generalize from the perspective of signal mixing and processing. Dimension Mixer model allows us to develop structured processing for unstructured data as well. Fig. 1 shows a simple example of a general Dimension Mixer model using two major operations (i) Select a group of dimensions and (ii) Process or Mix the selection. This operation is carried out in parallel as well as sequentially for efficient mixing of signals. The signal mixing can be evaluated with Graph Theory for the effective flow of signals.

The structure of Dimension Mixer model can be arbitrary in terms of input grouping/selection and mixing/processing function, as long as the input signals are efficiently mixed. The butterfly structure of the FFT is highly efficient, especially for GPUs. Hence, we focus on using the Butterfly Structure for its immediate utility to accelerate current Neural Network Architectures. We create more architectures using Dimension Mixer in Appendix D. Furthermore, we compare computational complexity of various architectures in Table 2.

Table 2: Comparison of computational complexity of various architectures. (LEFT) unstructured architecture with input-output dimension $n$. The complexity is measured for complete mixing in a single block containing maximum path-length layers. (RIGHT) structured architecture with sequence dimension ($s$) and patch/filter size ($k$). The complexity is measured for single layer of mixing and requires max path-length layers to fully communicate between all elements. Here, *Monarch* refers to *Butterfly* with only 2 factors. * represents our proposed models.

| Architecture | Complexity | Max Path-Length | Architecture | Complexity | Max Path-Length |
|---|---|---|---|---|---|
| Linear | $n^2$ | 1 | Attention | $s^2$ | 1 |
| Monarch Linear | $n\sqrt{n}$ | 2 | Monarch Attention* | $s\sqrt{s}$ | 2 |
| Butterfly Linear | $nlog(n)$ | $log(n)$ | Butterfly Attention* | $s$ | $log(s)$ |
| MLP | $n^2$ | 1 | Convolution | $s$ | $s/k$ |
| Monarch MLP* | $n\sqrt{n}$ | 2 | MLP-Mixer | $s$ | 2 |
| Butterfly MLP* | $nlog(n)$ | $log(n)$ | PatchOnly Mixer* | $s$ | $s/k_{max}$ |

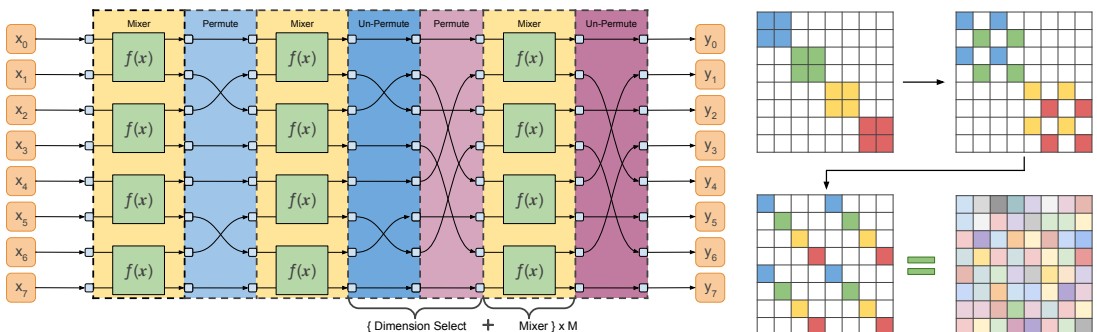

Figure 2: An example of FFT style Non-Linear Butterfly Mixer. This example shows the mixing of an 8-dimensional input signal using Radix-2 Butterfly. The first layer selects the dimension as it is. However, later layers use Permute to bring different dimensions in a block and later perform un-permute to place the dimension in their original location. For Radix-4 Butterfly a mixer block takes 4 dimensions as input and permutes accordingly as shown in Algorithm 2.

## 3.1. Non-Linear Butterfly Mixer

Although butterfly structure has been used widely in Linear Models [11–13], our theory of Dimension Mixer model allows non-linear mixing for effective mixing of input signals. We propose a non-linear butterfly structure model as shown in Fig. 2. The linear butterfly transformation is a special case with linear mixing function $f(x)$. Furthermore, we may use any learnable function as a mixing function and use butterfly structure to sparsify any dimension of input signal. This generalization allows us to extend butterfly structure beyond linear transforms. We can use MLP itself as non-linear mixing function to mix unstructured input dimension or Transformer as mixing function to mix long sequences. To this end, we propose Non-Linear Mixers called Butterfly MLP and Butterfly Attention.

**Butterfly MLP:** It uses MLP as a mixing function. We implement a column of MLPs in parallel which helps utilize the acceleration of GPUs. This simply breaks a whole MLP into blocks of MLPs in a butterfly structure for efficient mixing. Thus produced architecture is parameter efficient as well as allows having arbitrary MLP design (in terms of depth, width and activation) as a mixing function. Our method is more flexible than using Butterfly Linear as a replacement of dense layer. If we take the Radix-$N$ butterfly for $N$-dimensional input, it becomes a dense mixing. Hence, Butterfly MLP generalizes to standard MLP. However, Radix-$M$ ($M < N$) butterfly structure shards MLP into multiple smaller MLPs which is more parameter efficient, as well as highly parallelizable. The pseudocode for Butterfly MLP is shown in Appendix A. The generality of our mixing to multiple of block size is expanded in Appendix C.

**Butterfly Attention:** We also apply the efficient mixing of butterfly structures in the Attention mechanism of Transformers architecture (this is sparse but exact attention). It is widely known that

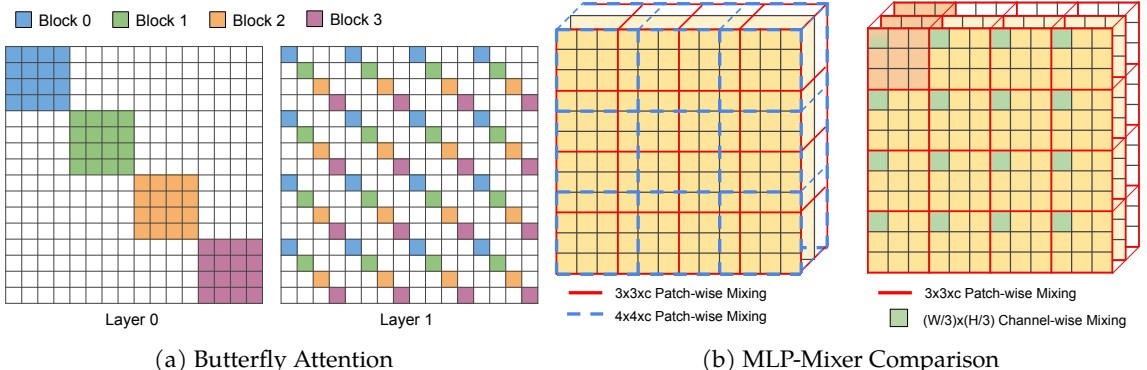

| Block 0 | Block 1 | Block 2 | Block 3 |
| --- | --- | --- | --- |

Layer 0      Layer 1

— 3x3xc Patch-wise Mixing
-- 4x4xc Patch-wise Mixing

— 3x3xc Patch-wise Mixing
▪ (W/3)x(H/3) Channel-wise Mixing

(a) Butterfly Attention      (b) MLP-Mixer Comparison

Figure 3: **(a)** An example of Butterfly Attention pattern on sequence length of 16 with butterfly structure of Radix-4. Using Radix-$\sqrt{N}$ creates two sparse attention matrix for complete mixing of signals. **(b)** (*left*) Patch Only MLP mixer (ours) compared to (*right*) Patch-Wise and Channel-Wise Mixing for 12x12 image size. The Channel-wise Mixing is replaced by Different size Patch-wise Mixing by our method.

---

**Algorithm 1** Permutation of Butterfly Attention for Transformers

---

```
# x: Input with shape [BATCH SIZE (B), SEQUENCE LENGTH (S), MODEL DIMENSION (D)]
# mask : Attention mask (binary) - EITHER: token-wise mask
# - OR: same size as Attention [-1, num_blocks, num_heads, block_size, block_size]
# block_size (a): Radix or Block size of Butterfly Attention
# i : Index of layer in butterfly (i.e 0, 1, .. log_a{S}-1 ; S is Sequence Length)
# transformer: a transformer layer with Attention and MLP layers [Vaswani et el.]
B, S, D = x.shape
for i, transformer in enumerate(transformers_layers):
    stride = block_size**i if ( block_size**(i+1) <= S ) else S//block_size

    x = x.view(B, -1, block_size, stride, D).transpose(2, 3).view(-1, block_size, D)
    x = transformer(x, mask)
    x = x.view(B, -1, stride, block_size, D).transpose(2, 3).view(B, S, D)
return x
```

---

the complexity of Attention is $S^2$ with sequence length $S$. Furthermore, there is a large number of cases where the sequence length can blow up significantly such as on large images when using small patch size, on long documents and on audio/video signals. To solve this issue, we apply partial block-wise attention using the butterfly structure, which reduces the complexity of Attention to just $S$, however it would require $\log_a(S)$ layers of Attention for complete mixing of tokens where $a$ is Radix or block size. For experiments, we use Radix-$\sqrt{S}$ for mixing which creates complete mixing within two attention blocks (see Fig. 3a). Since the MLP layer is invariant to permutation of tokens, we apply butterfly structure to Transformer Block as a whole as shown by the Algorithm 1. Since the Butterfly Attention converts Blocks of Tokens into batches, we may use any implementation of transformer architecture, including the optimized ones like Flash Attention [71].

Despite having sparse attention, experiments show that it performs better than full attention when used on CIFAR-10, CIFAR-100 datasets and on some LRA benchmark. Furthermore, the experiments show that the Butterfly Attention mechanism is faster for training as well as memory efficient.

## 3.2. Patch-Only MLP-Mixer for Vision

In the MLP-Mixer architecture, the mixing between two or more patches is done by token mixing (or channel-wise mixing) MLP. However, in CNN architecture, only use of patch-wise mixing is sufficient. This is due to the sliding window used in convolution which helps to mix the signals of

Table 3: Replacing Dense with Sparse layers in MLP-Mixer. The accuracy is calculated over 8 seeds.

| Method | Expansion | Parameters | MACs | Acc | Max-Acc | Permutations |
|---|---|---|---|---|---|---|
| Dense | | 351.68 k | 20.73 M | 83.83±0.407 | 84.64 | 0 |
| Butterfly-Linear | 1 | 140.96 k | 5.0 M | 82.45±0.361 | 83.01 | 4 |
| Butterfly-MLP | | 143.55 k | 5.33 M | 82.68±0.244 | 83.06 | 2 |
| Dense | | 615.29 k | 40.89 M | 84.07±0.355 | 84.47 | 0 |
| Butterfly-Linear | 2 | 193.86 k | 9.34 M | 83.49±0.322 | 83.84 | 4 |
| Butterfly-MLP | | 197.75 k | 9.88 M | 83.70±0.302 | 84.26 | 2 |

Table 4: An experiment comparing Butterfly Attention with Dense Attention. The B-Attention column represents if the Butterfly Attention is used. The Sequence Length for patch size 4, 2 and 1 are 64, 256 and 1024 respectively. Similarly, the embedding dimension for patch size 4, 2 and 1 are 128, 64 and 64 respectively. The device used for measuring time is GTX 3090 with PyTorch-2 and an off-the-shelf implementation of Transformers.

| Dataset | CIFAR-10 | | | | | | | | CIFAR-100 | | | |
|---|---|---|---|---|---|---|---|---|---|---|---|---|
| Layers, Patch | 8, 4 | | 4, 4 | | 4, 2 | | 4, 1 | | 4, 4 | | 4, 2 | |
| B-Attention | No | Yes | No | Yes | No | Yes | No | Yes | No | Yes | No | Yes |
| Accuracy (↑) | 81.90 | 84.69 | 81.04 | 83.68 | 78.35 | 80.80 | 65.65 | 73.96 | 53.92 | 57.02 | 50.03 | 53.29 |
| Time (ms) | 17.41 | 17.32 | 9.34 | 9.45 | 16.88 | 9.53 | 198.04 | 26.72 | 9.56 | 9.64 | 32.53 | 15.61 |
| Memory (MiB) | 434 | 386 | 304 | 270 | 1300 | 418 | 16592 | 1698 | 434 | 380 | 2468 | 810 |
| Params. (M) | 1.148 | | 0.618 | | 0.299 | | 0.79 | | 1.355 | | 1.773 | |

different regions along with an increase in the receptive field with an increase in depth. We search for a mechanism that allows for mixing using patch-only but without the sliding window.

We propose Patch-only MLP Mixer (Fig. 3b). The architecture consists of Image of size $I$ and Kernels/Patches of size $K_1$, $K_2$, $K_3$, $\cdots$ where, $I = K_1 \times K_2 \times K_3 \cdots$ such that $K_i$ and $K_j$ do not have common factors for different $i$ and $j$. If patch-sizes have common factors then the mixing occurs in different partitions corresponding to the factors without complete mixing. For example, if we have patches of size 6 and 8, then mixing happens in $lcm(6, 8) = 24$ instead of 48 block size. We choose only 2 factors for patch size for the simplicity of experiments. Patch-only MLP-Mixer lies in between MLP-Mixer and Convolution as it uses MLP for processing patches but only uses patches for overall input mixing.

## 4. Experiments

**Butterfly MLP:** We test the approximation capacity of (i) our Butterfly MLP, (ii) MLP with Butterfly-Linear Transform [13, 72] and (iii) Dense MLP in a MLP-Mixer settings trained on CIFAR-10 dataset. Both sparse models have large savings on parameters and compute. For the experiments, we use MLP-Mixer with 7 layers and train for 200 epochs with Adam optimizer(lr=0.001) and cosine decay lr. The dimension of patch and channel are 64 and 121 respectively - square numbers to create block size of 8 and 11 respectively. In Table 3, we compare model with hidden expansion of 1 and 2 on the MLP layer. Butterfly MLP produces slightly higher accuracy and has extra parameters consisting of biases in each mixing block of MLP unlike Butterfly Linear based MLP. The number of permutations (including the un-permute) used in MLP block shows that Butterfly-Linear requires 4 such permutations in 2 layers of sparse weights that follows MLP structure. However, Butterfly MLP accomplishes this with only 2 permutation and on a single butterfly structure. Here, a single block is itself non-linear and doesn't require two layers for approximating an MLP. Theoretically, this reduces data movement between parallel and sequential blocks.

**Butterfly Attention:** We test the capacity of Butterfly Attention as compared to dense Attention. We experiment on CIFAR-10 and CIFAR-100 datasets using different patch sizes and different number of layers without using Positional Encoding. In Table 4 we show the accuracy, parameters and memory

Table 5: Comparison of attention mechanisms on the Long Range Arena benchmark. The Standard attention [5] is a reference performance baseline. Among the sparse attention methods, Butterfly scores the best on most LRA tasks. Token length of a task is indicated in parentheses. *Legend: Flash-Attn. [71], Reformer [29], Linformer [30], Performer [31], Nystromfrm. [28]*

| Model | ListOps (2K) | Text (4K) | Retrieval (4K) | Image (1K) | Pathfinder (1K) | Average (≤4K) | Path-X (16K) |
|---|---|---|---|---|---|---|---|
| *Standard* | 37.10 | 65.02 | 79.35 | 38.20 | 74.16 | 58.77 | — |
| *Flash-Attn.* | 37.6 | 63.9 | 81.4 | 43.5 | 72.7 | 59.8 | 61.4 |
| Reformer | 19.05 | 64.88 | 78.64 | 43.29 | 69.36 | 55.04 | — |
| Linformer | **37.25** | 55.91 | 79.37 | 37.84 | 67.60 | 55.59 | — |
| Performer | 18.80 | 63.81 | 78.62 | 37.07 | 69.87 | 53.63 | — |
| Nystromfrm. | 37.15 | **65.52** | 79.56 | 41.58 | 70.94 | 58.95 | — |
| Butterfly | 37.05 | 65.25 | **81.32** | **44.02** | **71.12** | **59.75** | **76.72** |

usage of the vision transformers. The time taken per step is an average over 50 steps of training. The experiments are designed to test the resource consumption and accuracy of both sparse and dense attention models. Experiments show that Butterfly Attention scales better with longer sequence lengths – produces a faster, less memory consuming and better-performing model as compared to dense attention. We discuss the possible reasons for accuracy gains in Appendix B. The experiments are carried out for 300 epochs and 64 and 128 batch size for CIFAR-10 and CIFAR-100 respectively with cosine decay of learning rate 0.0001. We use a total of 8 Attention heads on all the models.

Table 6: Summary of Pathfinder datasets used to solve pathfinder-X (*Top to Bottom*). Accuracy is reported for Butterfly Attention based Transformer. Here, some of the *Pathfinder-32* and *Pathfinder-64* samples are scaled for reference.

| Image Size | Contour Length | Generated | Num Distractors | Paddle Gap | Paddle Contrast | Samples | Acc |
|---|---|---|---|---|---|---|---|
| 32 | 14 | ✗ | - | - | - |  | 69.17 |
| 64 | 9 | ✗ | - | - | - |  | 80.99 |
| 128 | 14 | ✓ | 5 | 1 | 0.9 |  | 71.84 |
| 128 | 14 | ✓ | 5 | 2 | 0.9 |  | 75.05 |
| 128 | 14 | ✓ | 14 | 2,3 | 0.8 |  | 75.46 |
| 128 | 14 | ✓ | 20 | 2,3 | 0.73 |  | 76.41 |
| 128 | 14 | ✗ | - | - | - |  | 76.72 |

**Long Range Arena:** We conduct experiments on the Long Range Arena (LRA) [63] benchmark to test the capacity of Butterfly Attention with regards to context length. The LRA benchmark is designed for a fair comparison of efficient attention mechanisms using the same training settings. The results on LRA along with their average accuracies are shown in Table 5. Experiments show that our efficient attention performs competitive among tasks with best average score. The data for other

Table 7: Vision MLP Mixers Comparision. We measure the Accuracy Parameters and MACs. MLP-Dims shows the dimension of two MLPs (m1 and m2) used in single block of mixing. The reported accuracy is best over 3 runs.

| Architecture | Layers | MLP-Dims m1,m2 | CIFAR-10 | | | CIFAR-100 | | |
|---|---|---|---|---|---|---|---|---|
| | | | Acc | Params | MACs | Acc | Params | MACs |
| MLP Mixer (c1) | | 81, 144 | 83.81 | 0.90M | 74.65M | 57.37 | 1.95M | 75.7M |
| Patch Only | 7 | 75, 147 | 84.66 | 0.81M | 23.04M | 55.55 | 1.14M | 23.37M |
| MLP Mixer (c2) | | 64, 153 | 84.16 | 0.88M | 60.48M | 58.38 | 1.77M | 61.36M |
| MLP Mixer (c1) | | 81, 144 | 83.03 | 1.23M | 106.36M | 56.34 | 2.28M | 107.41M |
| Patch Only | 10 | 75, 147 | 85.49 | 1.14M | 32.9M | 56.29 | 1.47M | 33.23M |
| MLP Mixer (c2) | | 64, 153 | 84.20 | 1.22M | 86.16M | 57.81 | 2.10M | 87.04M |

sparse and standard attention is from NystromFormer [28]. We use the same training and evaluation protocol for our methods as well. The experiments on LRA (except Pathfinder-X) consist of a simple Xformer architecture with 2 layers and 2 attention heads, learning rate of 0.0001 with warmup, and linear learning rate decay with 0.1 dropout for embedding, attention, MLPs, and residuals (refer to [28] for all hyperparameters).

***Solving Pathfinder-X:*** Until recently, the Pathfinder-X with sequence length of $128^2$ ($16K$) tokens was not solved. This problem helped ignite interest in model very long sequence length. Since many transformer papers use LRA and Pathfinder-X as a *model* performance benchmark instead of arena for *attention* variants, direct comparison is not possible. We include results for Flash-Attention [71] in Table 5 as it solves Path-X with dense attention. We modify a few hyperameters to solve the task – use 4 layers of butterfly transformer to allow 2 complete signal mixing using block size of $128$ with 4 attention heads. We use learning-rate of $3 \times 10^{-4}$ with cosine decay, weight-decay of $1 \times 10^{-3}$, and limit dropout to only embedding, attention-matrix and hidden units of MLP.

Since brute force approach to training Pathfinder-X (*Pathfinder128 contour-length-14*) did not work, we devised a step by step process to help the transformer grok the intermediate complexities. We follow upscaling method from [71], i.e. we use nearest neighbour interpolation of positional embedding and also increase the size of butterfly-block. Experimentally, we find it easier to grok *Pathfinder64 contour-length-9* than *Pathfinder64 contour-length-14* initialized from *Pathfinder32 contour-length-14*. To scale from *Pathfinder64 contour-length-9* to *Pathfinder128 contour-length-14* is a huge increase in complexity and makes it difficult to grok - it has many *distractors*, have varying *paddles/dash gap* and possibly lower *contrast*. We generate datasets with increasing complexity to the equivalent of *Pathfinder128 contour-length-14*. Table 6 summarizes the datasets used and accuracy achieved by our method.

**Vision MLP Mixers:** We compare our Patch-Only MLP mixer with the original MLP-Mixer architecture with a similar number of layers and a similar number of parameters in the mixing blocks. We try to balance the mlp dimension on both Mixer methods. However, due to different ways of scaling the input, the models do not have same parameters. The experiments show that our method produces comparative or even better results than the original MLP mixer on CIFAR-10 and CIFAR-100 datasets (32x32 images). For MLP Mixer config-1 (*c1*) we scale the input image to 36x36 size, extract patch of size 4 and expand channels by a factor of 3. Furthermore, for config-2 (*c2*) we use a patch size of 4 and expand channels by a factor of 3.2. On Patch-Only MLP Mixer, we expand the image to 35 = 5x7 size and mix over the patch of sizes 5 and 7 with no channel expansion. We train all models for 200 epochs with Adam optimizer(lr=0.001) and cosine decay lr. We use 64 batch size for CIFAR-10 and 128 for CIFAR-100.

We compare two methods with a similar number of parameters in the mixing blocks. The results in Table 7 show that our method performs competitively or even better than the original method. If we compare the speed of training these models, our method lags behind in wallclock time due to the Unfold and Fold operations needed to extract and combine patches. However, these methods do not count towards MACs. In our configuration, MLP-Mixer produces higher MACs than our method.

This is because MLP-Mixer produces large hidden image size, while our method produces smaller hidden image size and has a smaller classifier layer. Moreover, sparsity can be varied by selection of *patch size* and *hidden expansion* [73].

## 5. Conclusion

In this paper, we introduce a generic method of efficient input signal processing model called Dimension Mixer model. We employed our method on multiple host architectures, such as MLP and Attention Layers of the Transformer Architecture, thereby introducing sparsity; to be exact, butterfly sparsity. A yet another contribution we made in this paper is the introduction of Patch-Only MLP Mixer as an intermediate architecture between the original MLP-Mixer and the CNN.

All the proposed models are the application of the Dimension Mixer model which we find is insightful for analyzing the signal processing on deep learning architectures and also for designing newer models. Experimental results show that our proposed models are often more efficient and/or more accurate than the counterpart architectures.

*Limitation:* Our study mostly uses small scale datasets for comparison. Benchmarking with existing models on large scale datasets like ImageNet [74] can make our findings more significant.

## Acknowledgements

We thank Dr. Anton Obukhov for his help with running Pathfinder-X experiments on the Euler cluster of ETH Zürich.

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

## A. Butterfly MLP Algorithm

The pseudocode for Butterfly MLP is given by Algorithm 2.

---

**Algorithm 2** Implementation of Block-Sparse and Butterfly MLP

---

```
class BlockLinear:
    def __init__(self, num_blocks, input_block_dim, output_block_dim):
        self.weight = torch.randn(num_blocks, input_block_dim, output_block_dim)
        self.bias = torch.randn(num_blocks, 1, output_block_dim)

    def forward(self, x):
        ## x -> [num_blocks, batch_size, input_block_dim]
        return torch.batch_matmul(x, self.weight) + self.bias

class BlockMLP:
    def __init__(self, input_dim, layer_block_dims=[], actf=nn.GELU):
        self.block_dim = layer_dims[0]
        num_blocks = input_dim//layer_block_dims[0]

        ### Create a block MLP
        self.mlp = nn.Sequential([])
        for i in range(len(layer_block_dims)-1):
            self.mlp +=[ BlockLinear(num_blocks, layer_block_dims[i], layer_block_dims[i+1]),
                        actf() ]
        self.mlp = self.mlp[:-1]

    def forward(self, x):
        bs, input_dim = x.shape
        x = x.view(bs, -1, self.block_dim).transpose(0,1)
        x = self.mlp(x)
        x = x.transpose(1,0).view(bs, -1)
        return x

## x : Input with shape [batch_size, input_dim].
## block_dim -> size of block in block_sparse MLP. Usually a factor of input_dim
## block_layers : Layers of block mixing function.
## fn_block : Block mixing function; is parallel non-linear mixer operating per block.
## y : Output with same shape as Input x for simplicity.

block_layers = []
for _ in range(log_base(input_dim, base=block_dim)): ## using hidden expansion of 2
    block_layers += [ BlockMLP(input_dim, [block_dim, block_dim*2, block_dim]) ]

## Using Butterfly Permutation
for i, fn_block in enumerate(block_layers):
    stride = block_size**i if ( block_size**(i+1) <= input_dim ) else input_dim//block_size
    x = x.view(-1, block_dim, stride).transpose(2, 1).view(batch_size, -1)
    x = fn_block(x)
    x = x.view(-1, stride, block_dim).transpose(2, 1).view(batch_size, -1)
return x
```

---

## B. Effect of Patches, Block Size and Stride in Butterfly ViT

The use of patches as tokens in Butterfly ViT induces local effect on the division of blocks. This depends on the size of block to attend and the stride value to jump between tokens. This is depicted by the Figure 4. This structure might also help the sparse linear and non-linear MLP-Mixer architectures as shown in Table 3.

We randomize input patches to remove the inductive bias of locality in Butterfly ViT and only test for structured sparse mixing. We find that random tokens still perform well as

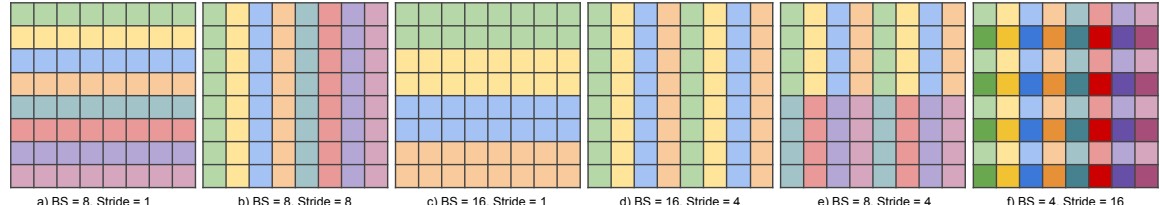

| a) BS = 8, Stride = 1 | b) BS = 8, Stride = 8 | c) BS = 16, Stride = 1 | d) BS = 16, Stride = 4 | e) BS = 8, Stride = 4 | f) BS = 4, Stride = 16 |

Figure 4: The images with 8x8 = 64 tokens using butterfly attention of different block size (BS) and various strides. Each different color represents different blocks of attention. The partial mixing of signals on multiple layers can create complete mixing of every tokens. Here, mixing combinations of $(ab)$, $(cd)$ $(cb)$, and $(cf)$ create complete mixing however, $(ae)$ and any same mixing like $(aa)$ does not mix every token.

shown in Table 8. These experiments do not use positional encoding, hence the butterfly provides even better inductive bias of the locality for processing the images or sequences.

To our surprise - using butterfly sparse attention with random token shuffling is still better than dense attention in some cases. We guess that sparse mixing itself can act as inductive bias.

## C. Permutation of Dimension

Previous works on Butterfly Structure of Linear Transform approximate matrices with power of 2 (*PoT*) [59] using $2 \times 2$ blocks or using resized *PoT* blocks [12]. Deformable Butterfly [72] gen-

Table 8: An experiment comparing Accuracy of ViT on CIFAR dataset with Structured Butterfly Attention, Randomization of Tokens on Butterfly Attention and Dense Attention. The Sequence Length for patch size 4, 2 and 1 are 64, 256 and 1024 respectively.

| Dataset | CIFAR-10 | | | | CIFAR-100 | |
|---|---|---|---|---|---|---|
| Layers, Patch | 8, 4 | 4, 4 | 4, 2 | 4, 1 | 4, 4 | 4, 2 |
| Butterfly | 84.69 | 83.68 | 80.80 | 73.96 | 57.02 | 53.29 |
| Rand Butt. | 82.41 | 81.86 | 77.41 | 70.65 | 54.96 | 49.31 |
| Dense | 81.90 | 81.04 | 78.35 | 65.65 | 53.92 | 50.03 |

eralizes this structure to more flexible sized matrices by using non-*PoT* butterfly factors while Monarch Butterfly [13] generalizes to two block-diagonal matrices. Our work generalizes butterfly structure to work on square matrices of *multiple of M* rather than *power of M* using only $M \times M$ blocks. Modern architectures use $N \to N$ transforms widely and on multiple devices. Hence it is significant to generalize butterfly structure that consider the size of the blocks.

Below, we show how our method of butterfly structure is different from Deformable Butterfly when fixing the block size. We use formulation from their work to show the difference.

*Recap on Deformable Butterfly (DeBut):* The authors define the notion of a real-valued DeBut factor as $R_{(r,s,t)}^{(p,q)} \in \mathbb{R}^{p \times q}$ that contains block matrices along its main diagonal, wherein each block matrix is further partitioned into $r \times s$ blocks of $t \times t$ diagonal matrices.

In essence, such densification flow can be generalized to deformable blocks arising from the product of two contiguous DeBut factors, one with diagonal sub-blocks ($t > 1$) and another with dense sub-blocks ($t = 1$), say $\mathbf{R}_{(r_2,s_2,t_2)}^{(p_2,q_2)} \mathbf{R}_{(r_1,s_1,1)}^{(p_1,q_1)}$ where $t2 > 1$. It can be further shown that such densifying product mandates $q_2 = p_1$ and $t_2 = r_1$, leading to:

$$R_{(r_2r_1,s_2s_1,1)}^{(p_2,q_1)} \leftarrow R_{(r_2,s_2,r_1)}^{(p_2,p_1)} R_{(r_1,s_1,1)}^{(p_1,q_1)}$$

*Edge case:* Let's try to decompose $8 \times 8$ matrix into two butterfly matrices.

We may take $q_1 = p_1 = q_2 = p_2 = 8$, $r_1 = s_1 = 4$, $r_2 = s_2 = 2$ and $t_2 = 4(= r_1)$ as per the definition. Here, the block-diagonal matrix have size of 4x4 and 2x2 respectively.

However, if we want to use both block-diagonal matrix of size $4 \times 4$, then we are limited by their statement that densifying mandates $t_2 = r_1$.

*With our method:* Our method does not follow the strict requirement stated previously that $t_2 = r_1$, but follows requirements that $q_2 = p_1$, $t_2 > 1$ and $t_2 \leq r_1$ leading to:

$$R^{(p_2,q_1)}_{(r_2 t_2, s_2 t_2, 1)} \leftarrow R^{(p_2,p_1)}_{(r_2,s_2,t_2)} R^{(p_1,q_1)}_{(r_1,s_1,1)}$$

Take $q_1 = p_1 = q_2 = p_2 = 8$, $r_1 = s_1 = r_2 = s_2 = 4$ and $t_2 = 2(\neq r_1)$. Here, we want to process $8 \times 8$ dimensions with block-diagonal mixers of size $4 \times 4$. This can approximate $8 \times 8$ matrix with two layers of $4 \times 4$ block diagonal matrices. We achieve this by using $\text{stride}(t_2) = q_2/r_1$ in the last layer as shown in Algorithm 1 and 2.

# D. More usage of Dimension Group Mixing

We show further use of partial or group mixing to create more parallelizable and efficient architecture.

## D.1. Token Parallel Attention

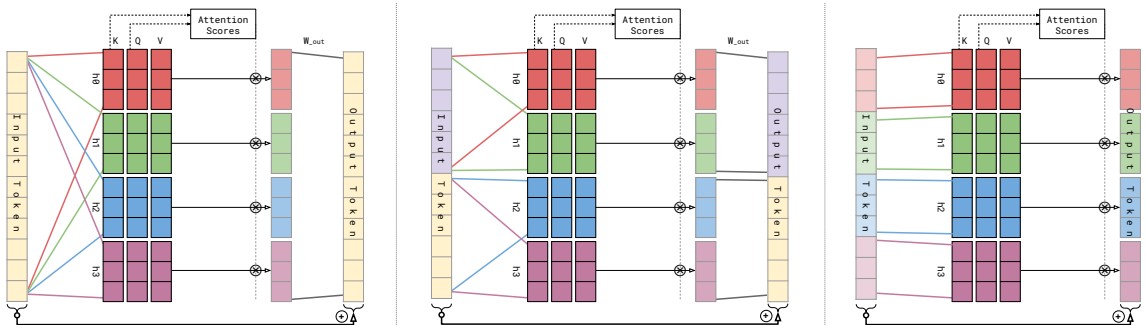

Figure 5: ($left$) The standard multi-headed self attention (MHSA). ($mid$) Token group parallel attention with token split into multiple parallel but smaller MHSA ($right$) Most reduced form that uses 1 head per token-group and token dimension is mixed by the next MLP layer. The attention heads in all figures are labeled as $h_i$.

Table 9: The number of heads is 8 for all experiments. Token-Parallel-2 has 4 parallel MultiHeaded Self-Attention with 2 heads per group. Token-Parallel 1 has 1 Self-Attention per group without using the $W_{out}$ matrix. The baseline is reported from Table 4. The reported accuracy is best among 3 runs.

| Layers, Patch
Token-Dims | 8, 4
128 | | 4, 4
128 | | 4, 2
64 | |
| --- | --- | --- | --- | --- | --- | --- |
| | Acc | Params | Acc | Params | Acc | Params |
| Baseline | 81.90 | 1.14 M | 81.04 | 0.618 M | 78.35 | 0.299 M |
| without $W_{out}$ | 81.02 | 1.06 M | 80.74 | 0.552 M | 77.89 | 0.282 M |
| Token-Parallel 2 | 81.57 | 0.721 M | 81.42 | 0.405 M | 76.95 | 0.245 M |
| Token-Parallel 1 | 82.18 | 0.689 M | 81.59 | 0.389 M | 76.83 | 0.241 M |

We focus on $W_{out}$ term of attention. First, according to our theory, its purpose is to mix multiple heads. It is clear that MLP layer just after attention can mix those signals. Can we simply remove that? The KQV matrices also take into consideration the whole of tokens. So, can we have block-sparse KQV matrices rather than low-rank for each heads in parallel? There has been some success with using split channel for reversible neural architectures [75, 76].

We experiment on ViT with 2 heads per parallel-attention-block as shown in Figure 5 and find reduced parameter and compute with not much degradation in performance. The experiment is performed on CIFAR-10 dataset as shown in Table 9

On top of that, if we have 1 head per parallel block, we can simply remove the $W_{out}$ matrix algebraically, as it can be combined with $W_v$ matrix. Experiments show that, it works fine and additionally helps fully parallelize heads in multi-headed self attention.

## D.2. Convolutional Channel Mixer

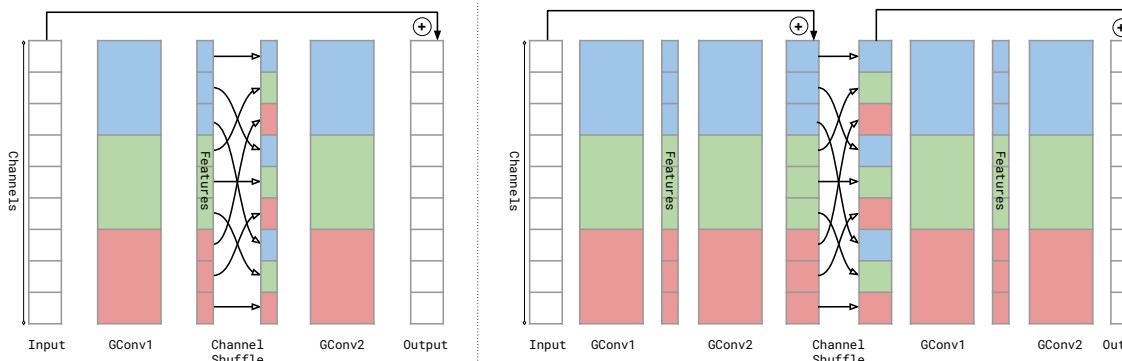

Figure 6: (LEFT) Residual block with Group Convolution and Channel Shuffle for mixing all signals in 1 Res-block. (RIGHT) Multiple grouped residual block with channel shuffle after residual block to mix channels using multiple residual-blocks. GConv1 layer consists of convolution, batchnorm and activation function, whereas GConv2 consists of convolution and batchnorm.

ShuffelNet [16] uses grouped convolution, channel shuffeling and grouped convolution in sequence to mix all channels. It focuses on sparse convnet using DepthWise [77] and PointWise convolution. For experimental comparision on CIFAR-10 dataset, we sparsify the convolutional channels of ResNet [4] architecture to mix within a single block as shown in Figure 6 (left).

One extension we make is to create parallel residual blocks grouped in channels, which do not mix all channels. We shuffle the groups and mix using next layer of parallel res-blocks. Extending this way helps us mix all signals with fewer shuffels/permutation for a constant number of layers as shown in Figure 6 (right).

With experiments as shown in Figure 7, we find that splitting the channels does not help much with sparsity-accuracy tradeoff. The results points to having accuracy dependent on parameters, and not improving due to sparsity. This could be because there is no structure (or inductive bias) to benefit the performance. This

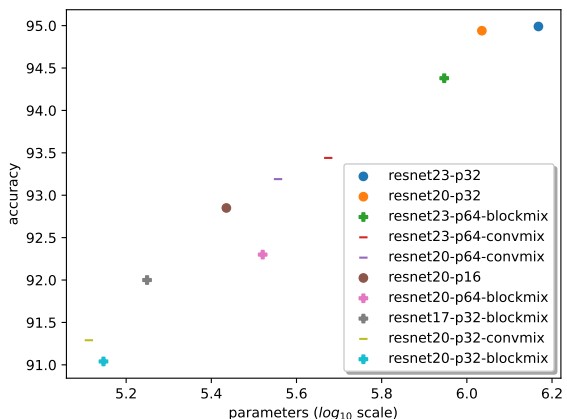

Figure 7: Result showing test accuracy of various ResNet modifications using different number of input-planes or channels. The legend shows resnet{depth}-p{input channels}-{mixing location}. conv-mix and block-mix is shown in Fig 6 left and right respectively. The reported accuracy is best over 3 runs.

approach might be helpful in creating parallel architectures, which can leverage fewer communication between blocks instead of communicating after each convolution operation.

