# OpenReview forum: "Dimension Mixer: Group Mixing of Input Dimensions for Efficient Function Approximation"
_CPAL.cc/2025/Proceedings_Track — CPAL 2025 (Proceedings Track) Poster_

### Official Review · Reviewer_bkQY · 2025-01-10
**Review for Dimension Mixer**

**Rating:** 6
**Confidence:** 4

**Review:**

**Summary:**

This paper introduce a dimension mixer model that try to improve the efficiency of the model architecture.


**Strongness:**

It's a good idea to split the dimension mixer model into two stages: select and mix. This design leads to more efficient architecture and reduces computational complexity since operations are performed on only portions of the input.

**Weakness:**

1. The connections to FFT and butterfly architecture is **misleading** for two main reasons:

    a. First, the outputs are not equivalent. While FFT is an accelerating algorithm that produces exactly the same output as DFT, the butterfly MLP's output cannot match a standard MLP's output.

    b. Second, the core concepts differ. FFT employs a divide-and-conquer approach, breaking DFT into smaller computations that can be efficiently combined. In contrast, butterfly MLP simply uses group-based permutation and computation. Its operations cannot be divided into subproblems, solved, and recombined.

    While both approaches involve grouping, I think there are no fundamental connections between their core concepts. If authors can justify other stronger connections, I'm willing to discuss.

2. The paper would benefit from comparing Big-O complexity between Butterfly MLP, Butterfly Attention, and Patch-Only MLP-Mixer versus standard MLP, Attention, and Convolution.

3. Regarding permutation patterns: The Butterfly MLP and Butterfly Attention use a fixed permute pattern—have you explored different permutation patterns? Would a random permute-unpermute approach work?

4. While ImageNet experiments may be resource-intensive, could you test the approach on ImageNet using smaller models like Tiny-DeiT? It's not about pursuiting SOTA results, but show effectiveness of the method on large-scale dataset.

---

### Official Review · Reviewer_EoaP · 2025-01-10
**Limited by small-scale experiments but interesting**

**Rating:** 7
**Confidence:** 3

**Review:**

This paper proposes "Dimension-Mixer," a general framework for mixing input dimensions in neural networks, inspired by FFT. It introduces the "Butterfly MLP," "Butterfly Attention," and "Patch-Only MLP Mixer." The core idea is to leverage a block-sparse butterfly structure for efficient mixing of input dimensions, reducing computational complexity while maintaining effective signal interactions.

Strengths:
The use of the butterfly structure is interesting and seems well-motivated for mixing dimensions sparsely but hierarchically, ensuring that all dimensions eventually interact. Block-sparse design appears well-aligned with the capabilities of modern accelerator hardware.
The paper is comprehensive, addressing multiple architecture components (e.g., MLPs, attention mechanisms) and demonstrating versatility across tasks like CIFAR, Pathfinder, and the Long Range Arena benchmarks.
The presentation is clear. The paper is well-structured and visual descriptions (e.g., Figures 4, 5) are effective.

Weaknesses:
The primary limitation is that experiments are restricted to small-scale datasets like CIFAR and Pathfinder. While the authors acknowledge this, it leaves the practical utility of the butterfly architectures in larger, real-world problems unexplored.
The method relies on heuristic choices (e.g., block sizes, patch configurations) that may require tuning, potentially complicating deployment across varied tasks.

Overall:
While the experimental scope is limited, the paper offers a conceptually interesting neural architecture with promising initial results. Validation on large-scale datasets would strengthen its contribution, but I'm okay with leaving this for future work.

---

### Official Review · Reviewer_hJnc · 2025-01-14
**Dimension mixer: a generalized framework for dimension mixing in inputs**

**Rating:** 5
**Confidence:** 3

**Review:**

The paper introduces Dimension Mixer, a generalized framework for efficient function approximation that leverages sparse, structured mixing of input dimensions. The overall goal is to achieve scalable and efficient function approximation while developing methods to sparsely mix input data across both structured and unstructured dimensions.

The authors reinterpret established neural architectures—such as CNNs, Transformers, and MLP-Mixers—as special cases of “select-and-mix” operations on input dimensions. Drawing inspiration from the Fast Fourier Transform (FFT), they extend this concept to nonlinear settings by introducing Butterfly MLP and Butterfly Attention for Transformers. Furthermore, they propose a Patch-Only MLP-Mixer for vision tasks, bridging the conceptual gap between MLP-Mixers and CNNs.

**Strengths:**
- The paper presents a unifying and well-motivated concept of dimension mixing, applicable across multiple neural architectures.
- The paper is tackling an important problem. Efficient sub-quadratic operations for large-scale signals (images, sequences) are quite relevant in deep learning, especially with the continued growth of model sizes.

**Weaknesses:**
- In Table 2, the performance difference between the Butterfly MLP and the Butterfly linear layer is marginal (smaller than the error ranges). It is not clear to me if this improvement stems from the nonlinear structure of the Butterfly MLP or from additional parameters such as biases. Some ablation studies here could help clarify the contributions of the individual components.
- As also noted by the authors, the evaluation results are primarily based on small datasets like CIFAR-10. Testing on larger datasets could provide a more comprehensive assessment of the benefits of Butterfly MLP compared to its linear counterpart.
- The writing and presentation in the paper can be improved.
    - In Section 3, the dimension mixing model is mentioned but not formally defined. Lines 157 and 176 refer to the "theory of dimension mixing", yet the explanation is limited to figures. In addition, terms like "partial mixing of signals" (line 122) are a little vague and could benefit from precise definitions.
    - The descriptions in Section 3.2 could be written more clearly to enhance reader comprehension.
    - Although the permute operation is detailed in Algorithm 2 (appendix), including an intuitive explanation in the main text would improve accessibility for readers.
    - A citation should be provided in line 45 for the sentence "While sparse parameterizations...."

---

### Meta-Review · Area_Chair_p5Hw · 2025-02-04

**Recommendation:** Accept (Poster)
**Confidence:** 4

**Metareview:**

This paper extends the idea of using structured sparse (butterfly) matrices in linear transforms to construct deep networks with structured sparse dimension mixing. This architecture breaks up input patches into blocks and only combines patches within a block to obtain a more efficient deep network. Since operations that combine patches are quadratic in the size of a block, breaking up the input into smaller blocks can help reduce the number of operations required.

The authors apply this principle to MLPs, transformers, and MLP-mixers in a range of classification tasks. Their results show that their approach is faster while achieving comparable accuracy. Reviewers note that Butterfly MLP results are not much better than Butterfly linear on CIFAR-10. Moreover the experiments on CIFAR-10/100 seem contrived (using a patch size of 1 pixel to demonstrate the largest efficiency gains). However the approach in the paper is still interesting and worth presenting at this conference in my opinion.

Including a comparison table of BigO complexity of different structured transformations as suggested by reviewer bkQY would add to the paper and the authors are expected to include this when they update the paper.

---

### Decision · Program_Chairs · 2025-02-11

Accept (Poster)